# Verifiable, Secure Mobile Agent Migration in Healthcare Systems Using a Polynomial-Based Threshold Secret Sharing Scheme with a Blowfish Algorithm

**DOI:** 10.3390/s22228620

**Published:** 2022-11-08

**Authors:** Pradeep Kumar, Kakoli Banerjee, Niraj Singhal, Ajay Kumar, Sita Rani, Raman Kumar, Cioca Adriana Lavinia

**Affiliations:** 1Department of Computer Science and Engineering, JSS Academy of Technical Education, Noida 201301, Uttar Pradesh, India; 2Institute of Engineering & Technology, Shobhit University, Meerut 250110, Uttar Pradesh, India; 3Department of Computer Science and Engineering, Chotu Ram Engineering College, Meerut 250001, Uttar Pradesh, India; 4Department of Computer Science and Engineering, Guru Nanak Dev Engineering College, Ludhiana 141006, Punjab, India; 5Department of Mechanical and Production Engineering, Guru Nanak Dev Engineering College, Ludhiana 141006, Punjab, India; 6CMI Cioca Adriana Lavinia, 22 Dorobantilor Street, 550231 Sibiu, Romania

**Keywords:** mobile agent migration, security, privacy, healthcare, threshold (k, n) secret sharing mechanism based on two polynomials, Blowfish

## Abstract

A mobile agent is a software application that moves naturally among hosts in a uniform and non-uniform environment; it starts with one host and then moves onto the next in order to divide data between clients. The mobile paradigm is utilized in a wide assortment of medical care applications such as the medical information of a patient, the recovery of clinical information, the incorporation of information pertaining to their wellbeing, dynamic help, telemedicine, obtaining clinical data, patient administration, and so on. The accompanying security issues have grown in tandem with the complexity and improvements in mobile agent technologies. As mobile agents work in an insecure environment, their security is a top priority when communicating and exchanging data and information. Data integrity, data confidentiality and authentication, on-repudiation, denial of service, and access control, are all key security concerns with mobile agent migration. This paper proposes a Verifiable, Secure Mobile Agent Migration model, based on two polynomials (t, n), and an edge secret imparting plan with Blowfish encryption, to enable secure information transmission in clinical medical care.

## 1. Introduction

Mobile agents [1] are innovations that originated in two distinct disciplines. The first discipline concerns artificial intelligence, and the idea that an intelligent agent is created [2]. The second discipline concerns distributive computing, wherein a mobile agent is created via code mobility. A legitimate definition for mobile agents, regarding the two referenced disciplines, is that they are smart programming substances that can pause and resume their tasks automatically, on various platforms, in order to complete assigned tasks [3]. A relocating mobile agent undertakes a process that is independent; it can navigate its way through, and adapt to, a heterogeneous environment, moving from platform to platform, and interfacing with different mobile agents. Mobile agents automatically [4] decide where and when to migrate, and they may execute their task at anytime, or they may suspend the execution of that task altogether, move to another host, and proceed with executing the task on that host instead. Characteristics of mobile agents are:Mobility: Mobile agents can freeze an operation on one platform and continue with the operation on another (i.e., inside a different region. This is often referred to as agent migration) [5].Individualism: Each mobile agent is guided by a program that is especially written to achieve at least one goal. The operations of mobile agents are entirely governed by this code, with no direct intervention from other groups.Reactivity: Mobile agents respond to environmental changes in order to accomplish their objectives.Proactivity: Mobile agents change their current circumstances and they take a few attempts to accomplish their objectives.Sociability refers to a mobile agent’s ability to interact with other mobile agents. This is important because some agents are only made aware of their present situation via communication with other agents.

The client–server paradigm and mobile agent paradigm are shown in Figure 1. Table 1 represents the difference between remote procedure calls and mobile agent technology.

Mobile agents roam around freely in a malicious environment; therefore, the chances of an attack on a mobile agent are high. Figure 2 shows the different types of conventional and new threat in mobile agent systems.

There are various advantages to the mobile agent paradigm shown in Figure 3. These include the fact that it is autonomous and self-driven, easily maintainable, fault-tolerant, parallel processing, dynamically adopted, it has a fever load on the network, there is less network delay, and it has a reduced compilation time.

### 1.1. Protection of Mobile Agents

A prime concern is the security of mobile agents [8]. The manner in which mobile agents work is shown in Figure 4. The security of mobile agents can be threatened via different dangerous assaults. Several types of attacks, including denial of service (DOS), masquerading, specialist and host cracking, renunciation, eavesdropping, information and data manipulation, and so on, are possible due to the mobile agents’ dynamic behavior.

Mobile agent technology suffers from security threats [9], which are divided into four main categories:Assault from agents on platforms;Agent-to-Agent Assaults;Assault from platforms on agents;Additional Assault to Agent Platform.

The challenges when implementing mobile agents include security risks [10], protection of hosts from malevolent specialists, protection of agents from noxious hosts, efficiency, flexibility, mobility, and standardization.

### 1.2. Mobile Agent Life-Cycle

The life-cycle of [7] the mobile agents (displayed in Figure 5) guarantees that they can adjust the climate (i.e., either at home or in an unfamiliar climate). They can switch between one hub and another, and they can hone in on their last result.

Creation: A new mobile agent is made, and the conditions of the mobile agent are initiated.Cloning: A specialist copy is made, and the present status of the first is duplicated in order to create cloning agents.Dispatch: A mobile agent moves to another host.Deactivation: The state of a mobile agent is saved in the repositories when it is in standby mode.Activation: The state of a deactivated mobile agent is restored from the repositories and applied to the lifetime mobile agent.Retraction: A mobile agent can converse with another agent and the stage.Disposal: The life-cycle of a mobile agent ends.Communication: Interactions between mobile agents and platforms.

## 2. Utilization of Mobile Agents in the Medical Care Domain

Mobile agents are used in a variety of medical-related activities [12], such as obtaining clinical information on executives, clinical data recovery, integrating information pertaining to a patient’s wellbeing, obtaining general clinical information, and so on. The use of mobile agents in the medical care sector ensures medical data integration [13] by combining information from different medical data sources, as shown in Figure 6.

Health Data Management: Acquiring, analyzing, and protecting medical information [14].Information Retrieval: Retrieving medical information from heterogeneous databases.Decision-Making Support: Assisting healthcare workers with procedures, including treatments and diagnostics.Telemedicine: Systems focused on remotely monitoring the situation with patients, thus allowing for a wide range of assessments.Securing Medical Information: Approaches to working, bearing in mind the wellbeing and security of patient information.

Figure 7 and Figure 8 represent the real-time application of mobile agents in healthcare.

### Requirement of Security

Security is a pressing issue for mobile agents [15] as they migrate from user to user. Security parameters are shown in Figure 9.

Confidentiality: Security of information and data (state + data + code) [8] among platforms and agents.Data integrity: Data and information [16] should not be interfered with by a third party.Availability: Data and information requested by the platform or agents should be easily accessible.

## 3. Related Work

Liu et al. [10] proposed a model based on the fusion of a virtual integrated clinical database system with the mobile agent paradigm. Mobile agents help send information between different medical clinics, and the proposed model was utilized to incorporate information from different clinical data frameworks. This model was seen as having exceptionally favourable uses for the following reasons: patient information was regularly, and immediately, obtained by the medical clinics, saving a great deal of time [4]; needless patient trials did not occur in some clinics, which saved time for crises; and it guaranteed secure and productive information sharing.

The framework was executed as follows:A patient arrived at the hospital, and the medical faculty requested the patient’s vital clinical records via a VI (Visualized Interface).A MAS (Mobile Agent Scheduler) [17] dispatched portable specialists to outside organizations and emergency clinics, in order to solicit and assemble data. When the portable specialist arrived at the outside organization, it finished the check assessment initiated by the MAS of the outside establishment, accumulated data from the outside foundation’s CIS (Clinical-information Index Storage), and it continued its journey to the organization.After visiting all of the required institutions, the mobile agents returned to the organization at which they were assigned. The list items were saved in the CIS and displayed via the VI, which allowed the specialists to make better decisions.

Burstein et al. [18] presented a design that consolidated the mobile agents’ ideas, and considered how they could dynamically help in the medical care crisis space. This paper explicitly demonstrates how to utilize mobile agents in order to help disorganized administrations. First, a crisis circumstance is considered, wherein a moderately aged man experiences cardiovascular failure. The onlookers promptly call for an emergency vehicle, noting the area and nature of the problem, any recognizable identification on the patient, and so forth. When on route to the site of the incident, paramedics dispatch pair of mobile agents. First, the mobile agents travel to the medical clinics close to the site of the incident in order to understand whether accessible specialists, attendants, beds, or offices will be required. The second mobile agent recovers the patient’s clinical history, as well as the attributes of the chosen emergency clinic. This ensures that when the paramedics arrive at the crisis site, their onboard terminal in the rescue vehicle now mirrors the nearby clinics’ terminals, which are ordered from the most to least helpful, in accordance with the situation.

Additionally, the paper noted how subtle clinical information pertaining to the patient, and subtle information concerning the local environment, are accessible. When heading to the chosen emergency clinic, a message is conveyed to the related emergency clinic specialist. When the rescue vehicle arrives at the medical clinic, this data lets the trauma center and health worker know how to plan for the appearance of a patient. This paper concludes by arguing that mobile agents are helpful in situations in which the accuracy of information is critical for a successful decision.

Orgun [19] focuses on a scenario wherein patients change medical clinics and have numerous episodes in different medical services offices, thus prompting patient-related data to be divided into different frameworks. The authors suggested an electronic medical agent model that uses various participating versatile specialists in order to effectively access, translate, learn, and take advantage of the data that is available on different wellbeing frameworks. This multi-agent framework, with a metaphysical component that is dependent on HL7, works with the accumulation of patient data across an entire medical service association. The electronic medical mgent model comprises different agents from different servers, including a merchant agent and a cosmology server. A server for agents provides an interface for the information. Applications are set up on the electronic medical agent model network, and they are composed of the aforementioned information in various configurations, with various field names, and the data is able to be compared with that of a similar patient. An agent merchant monitors every one of the agents in the framework at a random time, in addition to the library of the numerous partaking applications and datasets in the medical services association.

Chaouch et al. [20] analyzed mobile agent-based structures DiabMAS (Diabetes Multi-Agent System) for the remote clinical assessment of diabetic patients. The argument for deploying mobile agents is to reduce traffic by assigning agents to locations where activities can be completed and message transmissions are closed, thus reducing network stress. Here, the key idea is to screen and assure patients, thus diminishing costs with regard to the advancement of patient diseases, and furthermore, creating a movement that provides a moderate degree of satisfaction.

Hsu [21] focused on telemedicine, and they alluded to an application that couples the innovation of PC technology and correspondence with clinical advantages. It will be available in different forms: tele-education, tele-conference, tele-medical procedures, and tele-observing. Here, the principal impetus behind this project is to design a protected specialist that can implement telemedicine-based pair-to-pair organizing. The design of a pair-to-pair network based on the JXTA convention utilizes two models of telemedicine administration: unsurprising and erratic administrations. When a mobile agent holding patient data begins its journey by starting on one spot, before moving onto the next through the web, it tends to be assaulted by malevolent agents; therefore, the design utilizes a two-layer security component in order to answer agent-based telemedicine administrations for mobile agents. Security necessities require the design to be: non-renounceable, private, dependable, and consistent.

Pouyan [22] describes a three-layer agent-based paradigm for e-health care that consists of collecting, evaluating, controlling, and eventually conducting patient-related services via the internet in real-time. Patients, healthcare clinics, and the central hospital comprise the three tiers. Physicians, nurses, trainers, and representatives can be from any sector to form a dynamic virtual team. All of these actors are mobile agents. A specific agent that represents a patient’s description, circumstances, and health, assigns the patient to a specific member of healthcare staff. Some medication data is entered into a database in accordance with the to the responsibilities allocated to each member of healthcare staff. The diagnostics group sends the request to the central institution in order to obtain data from the data system.

Benachenhou [23] examined the utilization of mobile specialist innovations in terms of their ability to correspond between various distant areas. This is a particularly pertinent consideration for when a crisis happens, particularly in mass setback occurrences when a large number of casualties need clinical consideration. This paper examines the utilization of mobile agents for the quicker and exact recovery of information, when legitimate medical care hardware or the examination of clinical data is beyond the realm of possibility. The author gives a description of a crisis wherein a group of individuals is affected by eye contamination. The paper showed that scanning for, and enquiring into, expert care for every single case set aside time.

Indeed, the paper showed that if we assume every one of the specialists reaches out to experts by sending them the medical profiles of every single patient, tumult and blockage will unquestionably ensue, thus ensuring that not even one of them will receive help; however, if we assume that the experts contact every single patient, the correspondence traffic will be decreased, though, it will still take a great deal of time. The paper indicated that affected individuals will visit nearby medical service communities. Nearby clinical official investigations will receive data from patients that will be stored in the neighborhood framework. A unique clinical office will dispatch a specialist, or several specialists, to all servers in order to observe the aftereffects of essential tests on affected eyes. These officials will receive answers to their information requests.

Biswas [24] considered straight (t, n) secret sharing plans, and they noticed some of its benefits. The proposed (t, n) edge plot depended on Shamir’s SS scheme. The paper gives an itemized security investigation of every one of the three periods of the SS scheme, including the specific offer being made, secret key reproduction, and mystery key approval. In any case, two, rather than one, irregular polynomial plans for producing two offers for each member are utilized. In this plan, one polynomial is utilized to divide confidentiality between t or more members, and the other is utilized to approve the mystery key and to discover cheating, assuming that it happens. The plan does not utilize any stolen work that has been discovered; in any case, the coefficients of the two polynomials are connected, and thus, the offers made by them are connected. The plan utilizes two arbitrary polynomials, with one normal coefficient between them. For offers made as a result of cheating, both the polynomials are altered, and thus, (t-1) effective cheating methods carried out by deceptive members can be rendered unimportant. Hence, this plan is useful as it does not involve other conditions to discover cheating, and it includes additional calculations to check such conditions.

Ahmed El-Yahyaoui [25] introduced an encryption plot, wherein the clamor is steady and does not rely upon the homomorphic assessment of ciphertexts. The homomorphy of our plan is derived from basic lattice tasks (expansion and increase). In a cloud environment, the operating period of our cryptography aims for enhancement action, and it logs a request for a few milliseconds. The author proposed a novel homomorphic cryptography scheme that is indisputable. It consists of a symmetric, commotion-free, probabilistic encryption with a non-commutative ring quaternionic cipher text space. The proposed encryption has a viable application in terms of its ability to offer sound calculations, in an environment of scrambled information, during distributed computation; this can be applied to the protection of a large amount of information. It is an efficient and practical scheme, the safety of which hinges on resolving an over-characterized set of quadratic multidimensional polynomial requirements in a non-commutative ring. The two key aspects of our strategy are homomorphy and confidence. The proposed approach allows a remote, unreliable, networked computing system to conduct sophisticated computations over scrambled data while allowing the customer to verify the accuracy of their rethought actions throughout the decryption process.

Alexeis Garcia-Perez [26] notes the fact that as countries transition into a post-industrial world, where understanding the economy is characterized by radical innovations in the information technology area, the digital transformation of the health industry is crucial. In order to sustain sectoral growth and to fortify against its fragility by deploying the newest technologies, their use in the health and care ecosystem must be managed properly in terms of cyberresilience.

Patel [27] explored the principle objective of dissecting the presentation of these calculations using documents that had either a little or a large amount of information. The runtime and amount of storage used by these computations during the operation are factors to consider. The experimental results and graphic research show which computation is the most suitable for small and large datasets. Scientific outcomes portray the more reasonable calculation for time and memory imperative frameworks. Velibor Božić [28] underlines the difficulty of managing health risks. IT, usernames, and passwords are insufficient to describe manager risk assessment competencies. The incorporation of risk mitigation into medicine must be planned as a program with a multimodal team as the sector is significantly larger and more complicated.

Kakoli Banerjee [29] reported the goal of resolving issues within DNA. Sequence compression issues, a novel compression algorithm, was suggested in their work, and a different concept was put forth. This concept relates to subbands and specialized framework similarities. The findings of the mathematical calculations show that intra-chromosome matching produces a greater number of inter-chronological matches. In the present study, only singularly exact similarities are taken into account. For improved compression, this process can be extended to difficult matches. This approach can be used to condense DNA layouts effectively, and it can also be used to determine how close two different chromosomes of a comparable genome are to one another. 

These papers proposed a framework for the security of patient data and water information at a time of commutation using mobile agents. These are the new approach for the security of patient credentials and WQI, and it takes less time, bandwidth, and has a low latency compared with existing approaches. The proposed approach utilizes two polynomial-based secret sharing schemes with a blowfish algorithm.

## 4. Problem Statement

As traditional communication slows down when there is more congestion on the network, mobile agents play a vital role in quickly assessing a patient’s condition during a mass loss incident. Mobile agents manage to retrieve diagnostic data from a few heterogeneous sources of information and the data is presented to the customer upon request. Mobile agents also assist medical professionals when making decisions during treatment. Above all, the aim when developing mobile agents is to automate tasks with minimal human interference. As a smaller number of human assets is required, it is possible to use human assets for other clinical purposes. In clinical regions, the security of prosperity data is a crucial matter that is customarily taken care of in neighbourhood storage facilities.

When studying the literature available for mobile agents in medical care, we saw that mobile agents are broadly utilized in the recovery, transportation, and execution of clinical information. Clinical information [30] is profoundly confidential. Consequently, securing migrating agents is a crucial issue. The mobile agent needs to be navigated away from certain dangers, such as the divulgence of data, forswearing of administration, and defilement of data, and different types of protection need to be utilized in order to obtain a mobile agent. The conventional direction with regard to security continues, and the focal point remains to be the implementation of assurance systems inside the portable specialist. In any case, accentuation involves gradually pushing toward creating strategies that move toward agent security, which is a considerably more troublesome issue. The security of medical data, as well as the security of mobile agents, is a prime concern [31].

## 5. Proposed Solution

Preliminaries—The prerequisites of the developed framework are discussed in Section 5.1 and Section 5.2. The secret sharing scheme [32] and blowfish symmetric encryption [33] are the main components of the prerequisite of the proposed model.

### 5.1. Secret Sharing Scheme

The SS scheme is dependent on Shamir’s SS scheme; however, instead of one, two random polynomials, similar to the scheme in Liu et al. for generating two shares per participant, are used. The scheme does not use any cheating detection function [5]; however, the coefficients of the two polynomials are correlated so that the generated shares are related.

Details of the scheme:

The proposed (t, n) threshold SS scheme comprises three phases: share generation, secret key reconstruction, and secret key validation/cheating detection [34]. They are described below.

Share generation phase: In the secret share generation phase threshold scheme (t, n), a trusted dealer considers a random polynomial degree (t-1), as follows:(1)$$f\left(x\right)=a_{0}+a_{1}x^{1\;}+a_{2}x^{2\;}\ldots \ldots \ldots \ldots \ldots +a_{t-1}x^{t-1\;}$$
where $a_{0}$, $a_{1}$, ……,$a_{t-1}$ ≠ 0 € Zq for a large prime q.

The secret S € Zq is obtained from $f\left(x\right)$ by replacing (*x*) with 0 (i.e., s = $f\left(0\right)$ = $a_{0}$:). Another polynomial $f\left(x\right)$ of degree $\left(t-1\right),$ called the supporting polynomial, is also selected by the dealer, as follows:(2)$$g\left(x\right)=b+a_{1\;\;\;}x^{1\;}+b_{2\;\;\;}x^{2\;}\ldots \ldots \ldots \ldots \ldots +b_{t-1\;\;\;}x^{t-1\;}$$
where $b_{0}$, $b_{0}$, ……, $b_{t-1}$ ≠ 0 € Zq

such that the coefficients of polynomial $f\left(x\right)$ and $g\left(x\right)$ for 0≤i≤t−1, are random and arbitrary except for a single value of I, for which ai = bi; however, the value of i and the coefficient ai (and bi) are kept secret. In fact, these two polynomials are related implicitly, and in such a way that they can validate each other (i.e., if one polynomial is changed, the other is affected and vice versa).

The dealer generates n pairs of shares $\left(k,f\left(k\right),g\left(k\right)\right)\;for\;k=1\;to\;n$ using two polynomial equations, $f\left(x\right)$ and $g\left(x\right),$ respectively, and the dealer secretly distributes them to n participants. Upon secretly pooling their shares, the secrets can be reconstructed by any t share or more, as described below.

Secret key reconstruction phase: After secretly exchanging and receiving shares, any t of n shareholders can redesign the polynomial equation f(x) using the interpolation [35] formula, as follows:(3)$$f\left(x\right)=\underset{i=1\;to\;t}{\sum }f\left(i\right)\prod_{j=1to\;t\;j\neq i}\frac{x-j}{i-j}$$

Similarly, the polynomial g(x) is also reconstructed. Now, the secret s = $f\left(0\right)$ = $a_{0}$ is taken as valid and correct if the following phase is satisfied.

Secret key validation/cheating detection phase: For 0≤i≤t−1, the coefficients of $f^{\prime }\left(x\right)$ and $g^{\prime }\left(x\right)$ must satisfy ${a^{\prime }}_{i}$ ≠ ${b^{\prime }}_{i}$ except for a single value of I, for which ${a^{\prime }}_{i}$ = ${b^{\prime }}_{i}$, and where $f^{\prime }\left(x\right)$ and $g^{\prime }\left(x\right)$ are non-constructed polynomials.

### 5.2. Blowfish Encryption

The workings of the blowfish and DES-based encryption, based on the Feistel structure are as follows. Blowfish is a block cipher symmetric key cryptography encryption and decryption mechanism proposed by Bruce Schneier to replace the DES encryption and decryption approach. The manner in which the Feistel structure works is efficient and secure. It is one of the first secure cryptographic algorithms and is free to use.

Size of each block: In a blowfish symmetric algorithm, 64-bits of block size are used.Size of Symmetric key: The blowfish cipher used variable key length sizes, from 32 to 448 bits.Subkey: In a blowfish cipher eighteen subkey numbers were used for internal operations.Number of rounds used in the blowfish cipher: In the blowfish cipher, 16 rounds were used.Substitution boxes: There were four substitution boxes used in the blowfish cipher.

The complete encryption procedure of the blowfish algorithm is summarized as follows: the block diagram of the blowfish encryption algorithm is shown in Figure 10, and the block diagram showing blowfish decryption is shown in Figure 11.

**Step 1:** Creation of subkeys in the blowfish algorithm:In the blowfish algorithm for encryption and decryption operations, 18 subkeys are needed {P[0], P[1], P[2]...P[17]}, and the same subkeys are used in encryption and decryption.Eighteen subkeys are stored in eighteen P arrays, and each array consists of 32 bits.P[0] = “456f7d98”, P[1] = “55a788e4”………………. P[17] = “3434eb6d”The relationship between each subkey and input key has been changed, as follows:P[0] = Perform the XOR operation between P[0] and the first 32-bits of the applied input key.P[1] = Perform the XOR operation between P[1] and the second 32-bits of the applied input key.P[i] = Perform the XOR operation between P[i] and the (i + 1)th 32-bits of the applied input key.

(Rotate the key to the first 32 bits, based on its size.)

Perform the XOR operation between P[17] and the 18th 32-bits of the applied input key.

(Rotate the key to the first 32 bits, based on its size.)

P-arrays that result in 18 subkeys are used in the encryption procedure.

**Step 2:** Setting-up Substitution Boxes:

In the blowfish encryption and decryption process, substitution boxes (S-boxes) play a very important role. Every S-box has 256 entries, starting from S[i][0] to S[i][255], and each has an entry size of 32 bits.

**Step 3**: Encryption:From i = 1 to 16:Li = Li XOR Ri;Ri = F(Li) XOR Ri;Swap Li, Ri.Undo the previous exchange.R = Perform XOR between R and P17.L = Perform XOR between L and P18.To get 64-bit cypher text, combine L and R.

### 5.3. Proposed Framework

The framework concerning secure medical information transmission in healthcare is shown in Figure 12. The framework based on secure key generation using two polynomials, blowfish encryption, and a decryption algorithm works in two phases: the first phase pertains to secure key generation, and the second concerns encryption and decryption using the blowfish algorithm. In the second phase, all patient documents and reports are encrypted using the key generated in the first phase, in order to encrypt and decrypt with the blowfish algorithm. The same key is used to decrypt reports and information concerning a patient. A major advantage of the proposed approach is its ability to work well during times of emergency. Every hospital securely shares the information of patients with other hospitals; therefore, the doctor can start treatment without delay and save the patients’ lives.

#### 5.3.1. Threshold Secret Sharing Scheme Using Pair of a Polynomial Equations

Our SS scheme follows Shamir’s [31] scheme, in that the generation of participants’ shares, where an irregular polynomial $f\left(x\right)$ of degree t−1, is used with coefficients from Zq. In addition, another polynomial $g\left(x\right)$ is taken as a supporting polynomial of $f\left(x\right),$ such that all the coefficients are random, except for one coefficient that matches with a coefficient of $f\left(x\right).$ As a result, the shares of $\left(x\right)$ are associated with the shares of $g\left(x\right)$ and vice versa.The shares are also independent because a common coefficient does not provide any dependency between $f\left(x\right)$ and $g\left(x\right).$ Moreover, most t−1 dishonest participants with 2t−1  shares cannot derive $f\left(x\right)$ and $g\left(x\right)$ (and thus cannot get all shares) as two polynomials contain 2t−1  unknowns.If none of the shares are modified, any subgroup of ‘t’ participants can generate the correct polynomials for $f\left(x\right)$ and $g\left(x\right),$ as shown in Figure 13. Due to two random polynomials and a common coefficient between them, the probability of successful share modification is $\frac{1}{{\left(tq\right)}^{2}}$.

#### 5.3.2. Blowfish Encryption

The process begins with the generation of subkeys. A P-array is used to store these 18 subkeys, with each array element being a 32-bit item. Next, substitution boxes (S-boxes) [36] are required in both the encryption and decryption processes, with each S-box containing 256 entries S[i][0]...S[i][255], and each item being 32 bits. Then, the last step involves encryption, in which the following steps are carried out:From i = 1 to 16:Li = Li xor Ri;Ri = F(Li) xor Ri;Swap Li, Ri.Undo last swap.R = R xor P17.L = L xor P18.Concatenate L and R to obtain a 64-bit cipher text.

## 6. Implementation and Result

The security of mobile agents in the healthcare field is provided by combining two polynomial-based secret key schemes and a blowfish symmetric encryption algorithm. The proposed scheme was implemented using python programming, and it was compared with the secret sharing with CRT [37] and EULER [38] algorithms that use the same parameter. After analysis, it was observed that the turnaround time for secret generation and regeneration is much less than the CRT and EULER secret sharing schemes. Table 2 shows the hardware and software requirements for implementing the proposed framework. The average case analysis, regarding the turnaround times for the key generation and regeneration of the CRT, EULER, and two polynomial-based secret sharing schemes [39,40], is shown in Table 3. The results of Table 3 are presented in a graph in Figure 12. It was observed from Figure 14, that the total turnaround time for the two polynomial-based scheme, with regard to secret key generation is optimal compared with other mechanisms.

The best-case analysis of the turnaround times for key generation and regeneration with regard to the CRT, EULER, and two polynomial-based secret sharing schemes is shown in Table 4. The results of Table 4 are presented in the form of a graph in Figure 15. It was observed from Figure 15 that the total turnaround time for the two polynomial-based scheme, with regard to secret key generation, is optimal, as compared with other mechanisms.

The encryption and decryption times of any algorithm depend on the size of an input file. Table 5 shows the encryption and decryption times of different file sizes, which range from 100 to 1000 kb, and the time observed is noted in msec for the three algorithms: AES, DES, and the blowfish algorithm. The results shown in Figure 16 show that the blowfish algorithm is optimal compared with AES and DES; therefore, our proposed model for the encryption and decryption of patient reports and identity utilizes a blowfish symmetric encryption algorithm.

## 7. Conclusions

The security of patient personal information and information regarding disease is the main issue discussed in this paper. A mobile agent is used to migrate information from one healthcare center to another. The proposed model, based on the mobile agent, is utilized from a security-based perspective. Another issue facing mobile agent security is resolved by using the proposed model, which is based on two polynomial authentication systems. Here, the authors used the blowfish algorithm to encode a patient’s information. Blowfish is a symmetric cryptography algorithm, and the encryption and decryption of documents requires a secret key that is generated and authenticated by using two polynomial-based mechanisms. The reason behind using the blowfish algorithm is that it is optimal compared with DES and AES encryption and decryption algorithms. After analyzing the proposed model with other symmetric key cryptosystems, key generation, and the recreation of a key, was observed as being far better than the CRT and Euler approach. In future designs, the optimal secret key may be created and recreated, and a secret key based on threshold value can be developed, which may give better results than the proposed model.

## Figures and Tables

**Figure 1 sensors-22-08620-f001:**
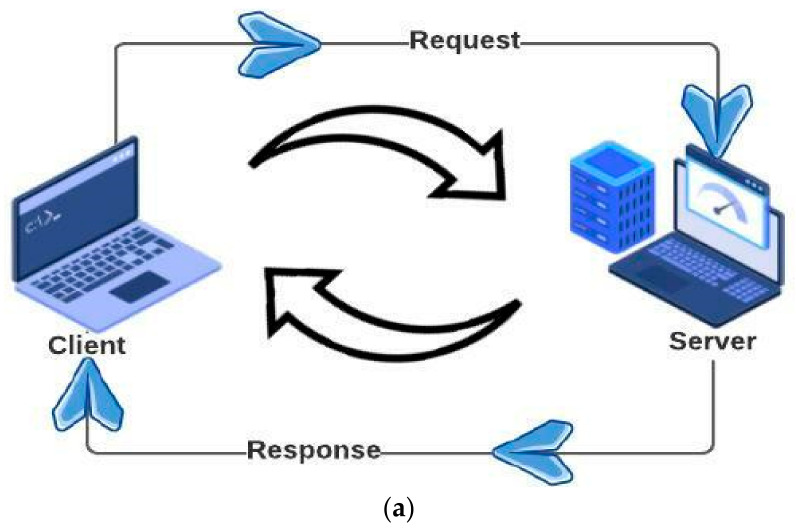
(**a**) Client–Server paradigm, (**b**) Mobile agent paradigm.

**Figure 2 sensors-22-08620-f002:**
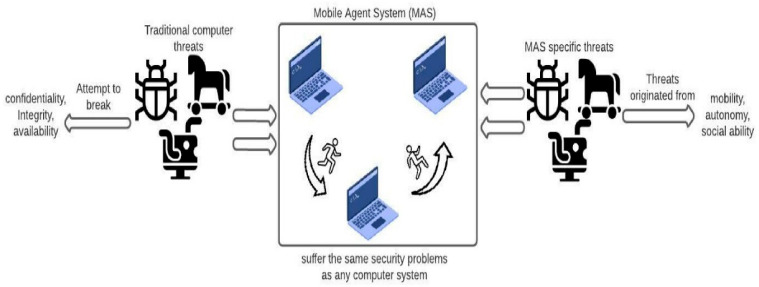
Conventional and new threats in MAS [7].

**Figure 3 sensors-22-08620-f003:**
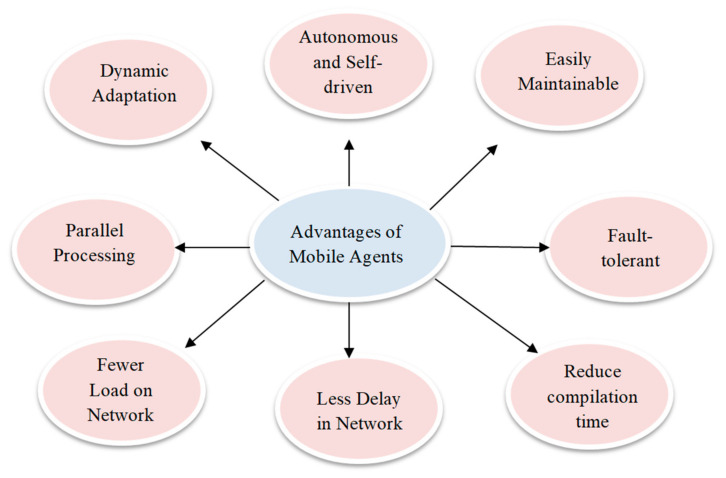
Advantages of mobile agents over conventional methods of protection.

**Figure 4 sensors-22-08620-f004:**
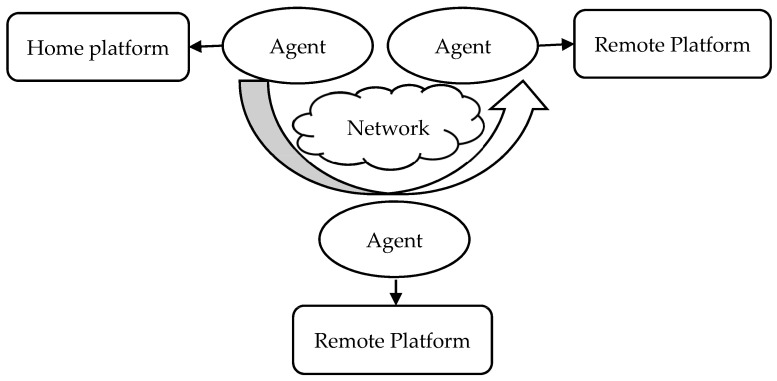
Main security threats to mobile agent technology.

**Figure 5 sensors-22-08620-f005:**
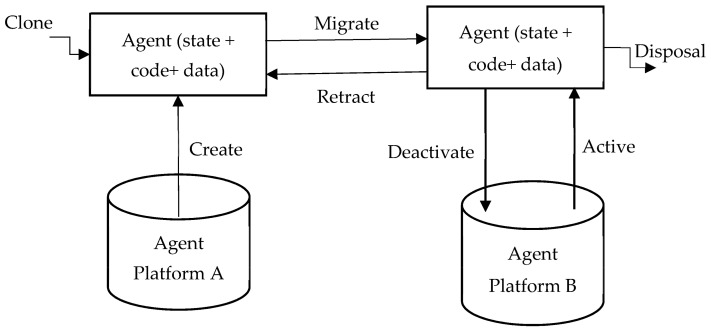
Mobile agent life-cycle [11].

**Figure 6 sensors-22-08620-f006:**
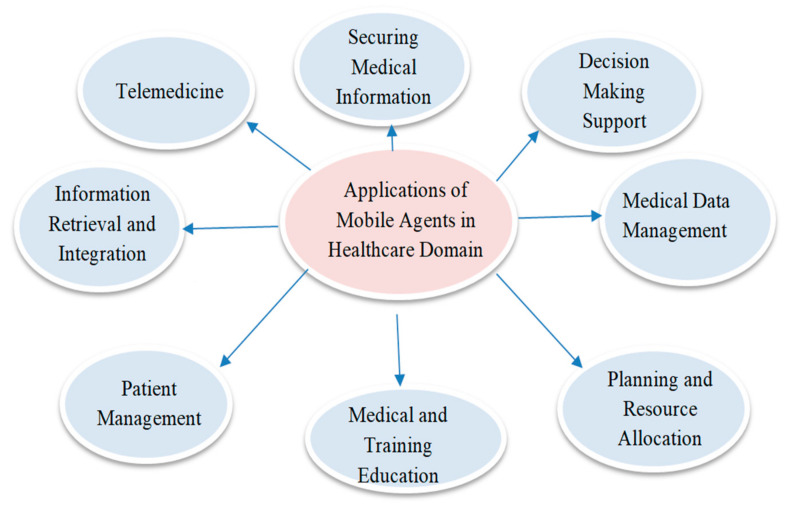
Applications of mobile agents in the healthcare domain.

**Figure 7 sensors-22-08620-f007:**
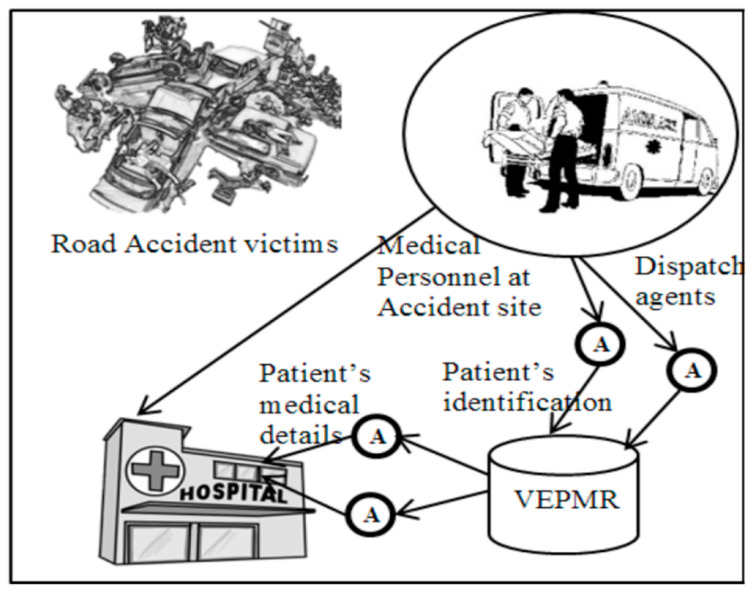
Application of a mobile agent in a road accident.

**Figure 8 sensors-22-08620-f008:**
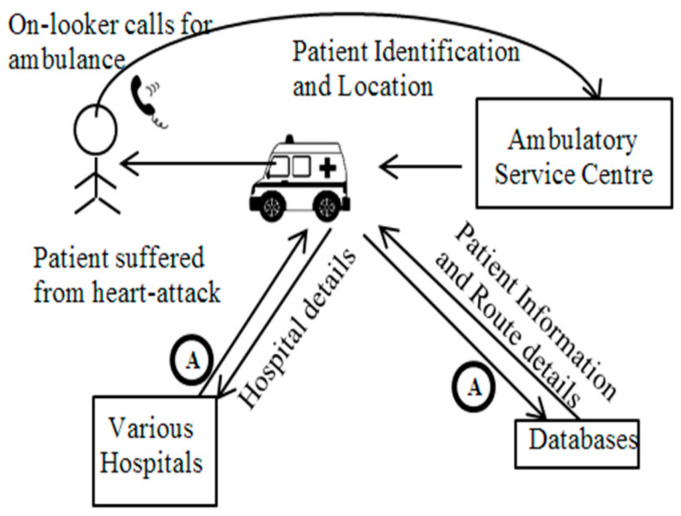
Application of a mobile agent in an emergency.

**Figure 9 sensors-22-08620-f009:**
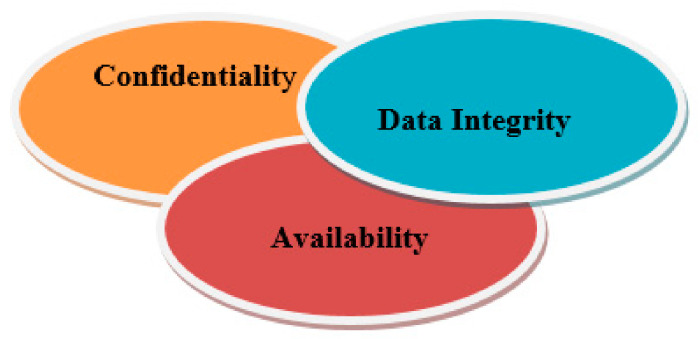
Security parameter.

**Figure 10 sensors-22-08620-f010:**
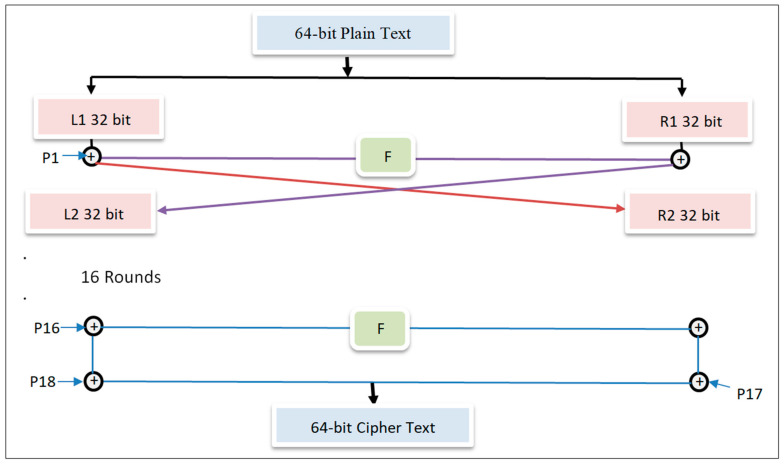
Blowfish encryption.

**Figure 11 sensors-22-08620-f011:**
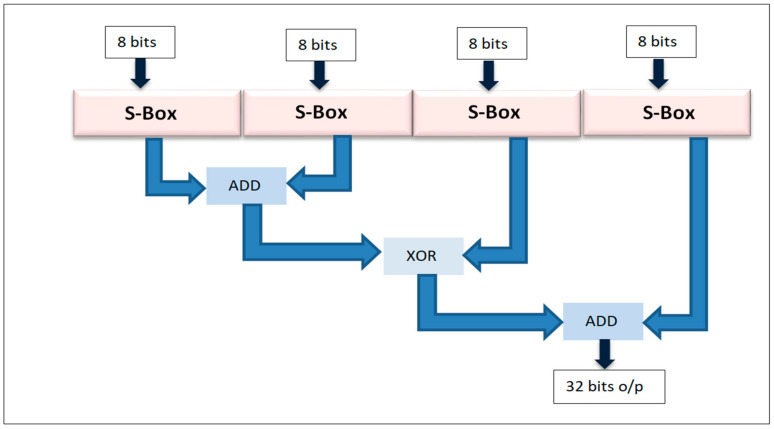
Function description.

**Figure 12 sensors-22-08620-f012:**
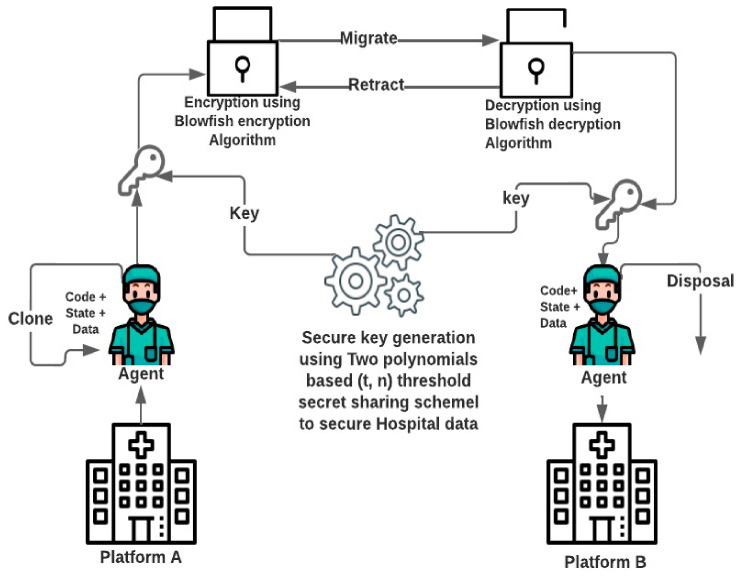
Secure agent migration framework in the healthcare system.

**Figure 13 sensors-22-08620-f013:**
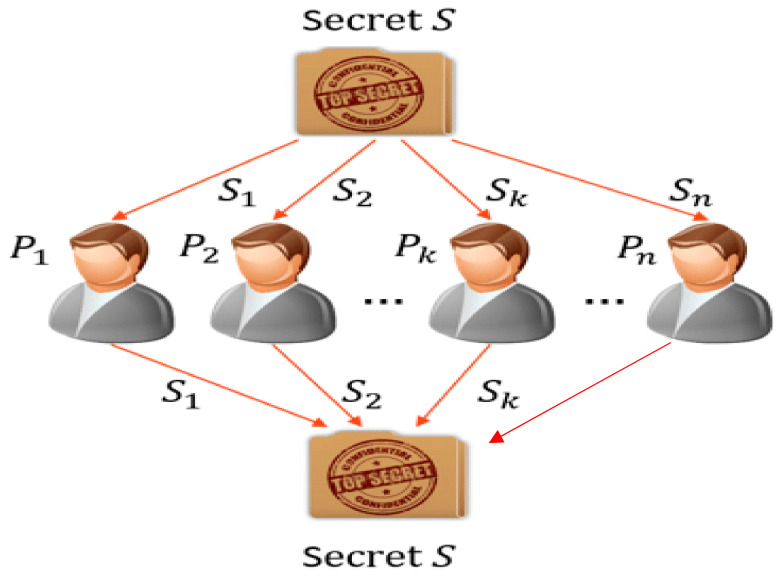
Secret sharing scheme.

**Figure 14 sensors-22-08620-f014:**
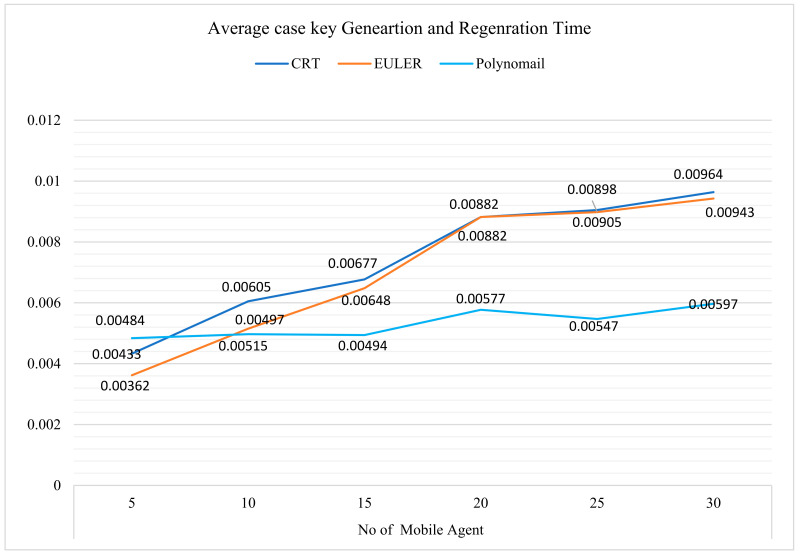
Comparison of average key generation and regeneration times.

**Figure 15 sensors-22-08620-f015:**
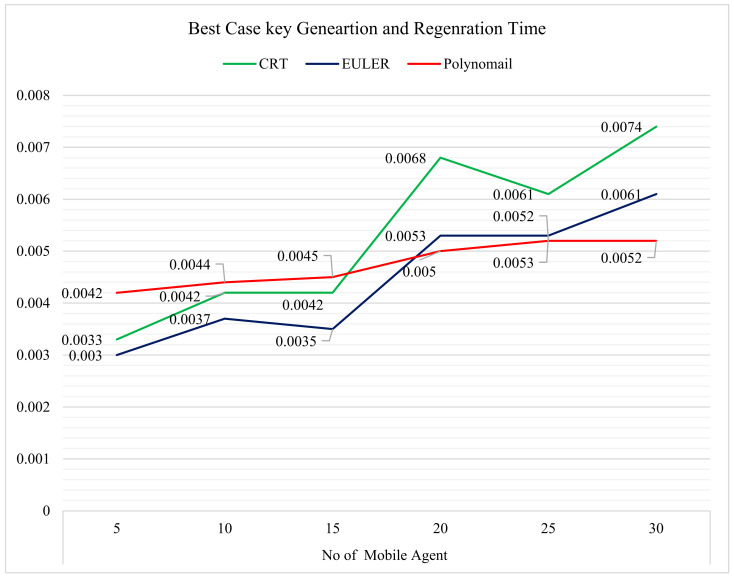
Comparison of best-case key generation and regeneration times.

**Figure 16 sensors-22-08620-f016:**
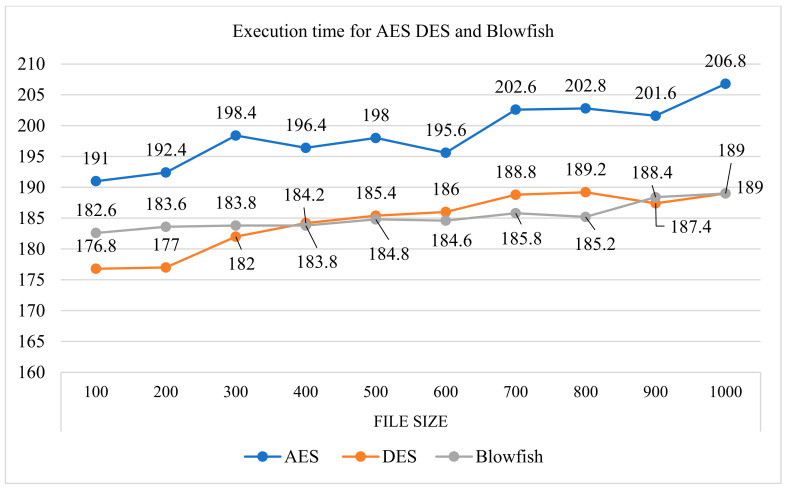
Execution times for the AES, DES, and blowfish algorithms.

**Table 1 sensors-22-08620-t001:** The differences between the conventional method RPC [6] and mobile agent technology [1].

Remote Procedure Call (RPC)	Mobile Agent Technology
Data transmission consumes greater channel capacity.	Rather than transmitting information, a process moves from one host to another, thus using less channel capacity.
The network load is high because many requests and replies are sent between the client and server.	The network load decreases because the main customer collaborator is a mobile agent, which moves to the server and in order to make neighbourhood associations.
They are not autonomous and self-driven.	They are autonomous and self-driven.
The user transmits the request to the computer as boundaries for a strategy. The strategy will be carried out over the server, with the results being returned to the user.	Mobile agents move from one host to the next and they work naturally.

**Table 2 sensors-22-08620-t002:** Requirements for computer configuration and software.

Name	Detail
Processor	Pentium IV
RAM	256 MB and above
Hard Disk	40 MB
Network	WAN
Software	Python

**Table 3 sensors-22-08620-t003:** Average case turnaround time for key generation and regeneration.

	No. of Mobile Agent					
	5	10	15	20	25	30
CRT	0.00433	0.00605	0.00677	0.00882	0.00905	0.00964
EULER	0.00362	0.00515	0.00648	0.00882	0.00898	0.00943
Polynomial	0.00484	0.00497	0.00494	0.00577	0.00547	0.00597

**Table 4 sensors-22-08620-t004:** Best case turnaround time for key generation and regeneration.

	No. of Mobile Agent					
	5	10	15	20	25	30
CRT	0.0033	0.0042	0.0042	0.0068	0.0061	0.0074
EULER	0.003	0.0037	0.0035	0.0053	0.0053	0.0061
Polynomial	0.0042	0.0044	0.0045	0.005	0.0052	0.0052

**Table 5 sensors-22-08620-t005:** Total time taken for encryption and decryption using the AES, DES, and blowfish algorithms.

File Size	100 MB	200 MB	300 MB	400 MB	500 MB	600 MB	700 MB	800 MB	900 MB	1000 MB
AES	191	192.4	198.4	196.4	198	195.6	202.6	202.8	201.6	206.8
DES	176.8	177	182	184.2	185.4	186	188.8	189.2	187.4	189
Blowfish	182.6	183.6	183.8	183.8	184.8	184.6	185.8	185.2	188.4	189

## Data Availability

Not applicable.

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
