# Peer review of "Verifiable, Secure Mobile Agent Migration in Healthcare Systems Using a Polynomial-Based Threshold Secret Sharing Scheme with a Blowfish Algorithm"

_sensors, 2022, doi:10.3390/s22228620_

Round 1
Reviewer 1 Report
1. The paper is interesting and dicusses the social problem of healthcare; 2. it is a good idea to add some photos of measurements; 3. it is a good idea to add a block diagram of the proposed research; 4. please include the recent papers as literature in reference. 5. please add photos of the application of the proposed research, if possible 2-3 photos; 6. figures should have high quality; 7. also highlight the benefits to society from this paper?", 8. compare your approach with other approaches advantages/disadvantages 9. please add some sentences about future work; 10 .Conclusion: point out what have you done;Author Response
Review Report (Reviewer 1)
Revised Submission of Manuscript (sensors-1955300)
Comments and Suggestions for Authors
Response: The authors earnestly thank the reviewers for pointing out the shortcomings of the manuscript. The comments have helped us to improve the quality of the manuscript further. All changes done are highlighted in the manuscript.
Comment 1. The paper is interesting and discusses the social problem of healthcare;
Response: Thanks for the inspirational words.
Comment 2. it is a good idea to add some photos of measurements;
Response: It is a good suggestion; now, Photos of measurements are included in the paper on pages 14, 15, 16 and 17.
Comment 3. it is a good idea to add a block diagram of the proposed research;
Response: The proposed model is shown in Figure 10 on page 13.
Comment 4. please include the recent papers as literature in reference.
Response: Thanks for the advice; now, recent paper references are included; Ref. No. 28, 31, 38
Comment 5. please add photos of the application of the proposed research, if possible 2-3 photos;
Response: The application of mobile agents in health care is shown in Figure 6 on page 5.
Comment 6. figures should have high quality;
Response: All Figures have been checked for quality standards, and quality enhancement has been completed.
Comment 7. also highlight the benefits to society from this paper?”,
Response: Benefit shown on page number 12. Figure 7 and Figure 8.
Comment 8. compare your approach with other approaches advantages/disadvantages
Response: Comparison of Proposed work shown in implementation and results.
Comment 9. please add some sentences about future work;
Response: Done in conclusion and Future work section,
Comment 10. Conclusion: point out what you have done;
Response: Done in the Conclusion section
We hope you will give full consideration to our submission.
Regards
All authors

Reviewer 2 Report
Comments:
This paper discussed the security of Patient personal information and information regarding diseases based on the mobile agent. A two polynomial authentication system or algorithm is used to resolve mobile agent security. This paper is very interesting and well written, but seems like the topic is not suitable for the section of Biomedical sensors in the SENSORS journal. In another word, this paper has nothing to do with sensors. Some of comments are shown following:
1. Page 1, line 27, check the expression: clinical information the executives.
2. Page 4, line 82, please give the full expression of the abbreviation “DOS”.
3. Figure 4. Check the caption. It is better to change it into “Main security threats to Mobile agent technology”
4. Page 14, line 429, please give the computer info, such as type, RAM, processor and so on.
Author Response
Review Report (Reviewer 2)
Revised Submission of Manuscript (sensors-1955300)
Comments and Suggestions for Authors
This paper discussed the security of Patient personal information and information regarding diseases based on the mobile agent. A two polynomial authentication system or algorithm is used to resolve mobile agent security. This paper is very interesting and well-written, but it seems like the topic is not suitable for the section of Biomedical sensors in the SENSORS journal. In other words, this paper has nothing to do with sensors. Some of the comments are shown following:
Response: The authors earnestly thank the reviewers for pointing out the shortcomings of the manuscript. The comments have helped us to improve the quality of the manuscript further. All changes done are highlighted in the manuscript.
Comment 1. Page 1, line 27, check the expression: clinical information the executives.
Response: Thanks for the beautiful observation; now corrected as “medical information of patient”.
Comment 2. Page 4, line 82, please give the full expression of the abbreviation “DOS”.
Response: Denial of Service (DOS).
Comment 3. Figure 4. Check the caption. It is better to change it to “Main security threats to Mobile agent technology”
Response: Thanks for the suggestion; now the Figure 4 caption has changed.
Comment 4. Page 14, line 429, please give the computer info, such as type, RAM, processor and so on.
Response: The required information is added to Table 2. Requirement of computer configuration and Software
We hope you will give full consideration to our submission.
Regards
All authors

Reviewer 3 Report
Verifiable Securing Mobile Agents Migration in Healthcare 2 System using Polynomial-based Threshold Secret Sharing 3 Scheme with Blowfish Algorithm is proposed in this manuscript. We think this manuscript is well organized and well written. In the review process I have following comments.
* The introduction section should describe the advantages and benefits of the proposed method. The Motivation and Contribution section should be integrated with the introduction.
* The quality of some figures is low and should be redesigned.
* In "Related works" section should focus more on differences between this paper and other works to highlight the novelty of this paper. Also the disadvantages and shortcomings of the previous methods that are addressed in the proposed method must be stated.
* The proposed method has no proper structure and is not well described.
* Some references are out-of-date, so these references before 2000 should be deleted. At the same time, many important recent references are missing, which can support the idea of this paper, the following references should be added in the Section "Related Work":
"A secure three-factor authentication scheme for IoT environments", Journal of Parallel and Distributed Computing, 2022, DOI: 10.1016/j.jpdc.2022.06.011
Author Response
Review Report (Reviewer 3)
Revised Submission of Manuscript (sensors-1955300)
Comments and Suggestions for Authors
Verifiable Securing Mobile Agents Migration in Healthcare 2 System using Polynomial-based Threshold Secret Sharing 3 Scheme with Blowfish Algorithm is proposed in this manuscript. We think this manuscript is well organized and well written. In the review process, I have following comments.
Response: The authors earnestly thank the reviewers for pointing out the shortcomings of the manuscript. The comments have helped us to improve the quality of the manuscript further. All changes are highlighted in the manuscript.
Comment 1. The introduction section should describe the advantages and benefits of the proposed method. The Motivation and Contribution section should be integrated with the introduction. The quality of some figures is low and should be redesigned.
Response: Thanks for this suggestion, as advised, as changes are done and highlighted. Figures have been redesigned.
Figure 4: Mobile Agent paradigm
Figure 5: Mobile Agent Life cycle [11]
Comment 2. In “Related works” section should focus more on differences between this paper and other works to highlight the novelty of this paper. Also the disadvantages and shortcomings of the previous methods that are addressed in the proposed method must be stated.
Response: Thanks for these suggestions and all changes done on page number 9.
Comment 3. The proposed method has no proper structure and is not well described.
Response: Proposed method is shown in Figure 10 on page number 13.
Comment 4. Some references are out-of-date, so these references before 2000 should be deleted. At the same time, many important recent references are missing, which can support the idea of this paper; the following references should be added in the Section “Related Work”:
Response: As advised, all changes have been incorporated into the paper.
We hope you will give full consideration to our submission.
Regards
All authors

Round 2
Reviewer 2 Report
The revised manuscript is good. Thanks for your efforts.